# A Multi-Modal and Multitask Benchmark in the Clinical Domain

## Abstract

Healthcare represents one of the most promising application areas for machine learning algorithms, including modern methods based on deep learning. Modern deep learning algorithms perform best on large datasets and on unstructured modalities such as text or image data; advances in deep learning have often been driven by the availability of such large datasets. Here, we introduce Multi-Modal Multi-Task MIMIC-III (M3) — a dataset and benchmark for evaluating machine learning algorithms in the healthcare domain. This dataset contains multi-modal patient data collected from intensive care units — including physiological time series, clinical notes, ECG waveforms, and tabular inputs — and defines six clinical tasks — including predicting mortality, decompensation, readmission, and other outcomes — which serve as benchmarks for comparing algorithms. We introduce new multi-modal and multi-task models for this dataset, and show that they outperform previous state-of-the-art results that only rely on a subset of all tasks and modalities. This highlights the potential of multi-task and multi-modal learning to improve the performance of algorithms in the healthcare domain. More generally, we envision M3 as a general resource that will help accelerate research in applying machine learning to healthcare.

## 1 Introduction

Healthcare and medicine are the some of the most promising areas in which machine learning algorithms can have an impact (Yu et al., 2018). Techniques relying on machine learning have found successful applications in dermatology, ophthalmology, and many other fields of medicine (Esteva et al., 2017; Gulshan et al., 2016; Hannun et al., 2019).

Modern machine learning techniques — including algorithms based on deep learning — perform best on large datasets and on unstructured inputs, such as text, images, and other forms of raw signal data (You et al., 2016; Agrawal et al., 2016). Progress in modern machine learning has in large part been driven by the availability of these types of large datasets as well as by competitive benchmarks on which algorithms are evaluated (Deng et al., 2009; Lin et al., 2014).

Recently, machine learning algorithms that combine data from multiple domains and that are trained to simultaneously solve a large number of tasks have achieved performance gains in domains such as machine translation and drug discovery (Johnson et al., 2017; Ramsundar et al., 2015). Current research in this area is driven by widely adopted computational benchmarks, particularly in the field of natural language processing (Wang et al., 2018a; 2019).

In this paper, we argue that multi-modal and multitask benchmarks can similarly drive progress in applications of machine learning to healthcare. In many healthcare settings, we have access to data coming from diverse modalities — including radiology images, clinical notes, wearable sensor data, and others — and we are solving many tasks — for example, estimating disease risk, predicting readmission, and forecasting decompensation events. These kinds of settings are naturally suited to modern deep learning algorithms; developing models that effectively leverage diverse tasks and modalities has the potential to greatly improve the performance of machine learning algorithms in the clinical domain.

As a first step in this research direction, we introduce in this paper Multi-Modal Multi-Task MIMIC-III (M3)[1], a dataset and benchmark for evaluating machine learning algorithms in healthcare that is inspired by popular multitask benchmarks in other application domains, such as natural language processing (Wang et al., 2018b; McCann et al., 2018). Previous clinical datasets and benchmarks have either focused on specific tasks in isolation as in Khadanga et al. (2020) or on multiple tasks over a single input modality (Harutyunyan et al., 2019). Our work is the first to combine multiple tasks and modalities into one benchmark.

More specifically, we propose a dataset that is derived from the MIMIC-III database and is comprised of data collected from over forty thousand patients who stayed in intensive care units (ICUs) of the Beth Israel Deaconess Medical Center between 2001 and 2012 (Johnson et al., 2016). As part of this dataset, we have collected data from four modalities — including physiological time series, clinical notes, ECG waveforms, and tabular data — and have defined six clinical tasks — mortality prediction, decompensation, readmission, and others. We also propose an evaluation framework to benchmark models on this dataset.

As a demonstration of how the M3 benchmark can drive progress in clinical applications of machine learning, we propose a first set of multi-modal and multitask models and evaluate them on our new benchmark. We find that these models achieve high performance levels and may serve as strong baselines for future work. In particular, our models outperform previous state-of-the-art results that only rely on a subset of all tasks and modalities.

These results highlight the potential of multitask and multi-modal learning to improve the performance of algorithms in the healthcare domain. We envision M3 as a general resource that will help accelerate research in applying machine learning to healthcare. To facilitate such uses, we release M3 and our models as an easy-to-use open-source package for the research community.

**Contributions.** In summary, our paper makes the following contributions.

- We define a new benchmark for machine learning algorithms in the clinical domain. It defines six clinical tasks, and is the first to collect data across multiple modalities.

- We introduce new multi-modal and multitask machine learning models which outperform previous state-of-the-art methods that only rely on a subset of tasks or modalities. This highlights the importance of multi-modal and multitask learning in clinical settings.

- We package our benchmark into an easy to use format such that the clinical machine learning community can further build upon our work.

## 2 BACKGROUND

**Machine Learning in the Clinical Domain.** Machine learning has been successfully applied throughout healthcare, including in areas such as medical imaging, drug discovery, and many others (Rajpurkar et al., 2017; Vamathevan et al., 2019). In this paper, we restrict our attention to a specific healthcare setting — intensive care.

The Medical Information Mart for Intensive Care (MIMIC-III) database is one of the most important resources for applying machine learning to intensive care (Johnson et al., 2016). Data collected in the ICUs includes vital signs, lab events, medical interventions, and socio-demographic information.

**Multi-Modal and Multi-Task Learning.** Multitask learning trains models to simultaneously solve multiple tasks (Ruder, 2017). Successful applications of multitask learning include machine translation and drug discovery (Johnson et al., 2017; Ramsundar et al., 2015). Current research in this area is driven by popular benchmarks, particularly in the field of natural language processing (Wang et al., 2018b; 2019; Rajpurkar et al., 2016).

Multi-modal machine learning combines and models data of different modalities such as vision, language, speech. A key challenge in multi-modal learning is to combine representations over diverse input types. Applications of multi-modal learning include image captioning and visual question answering (Anderson et al., 2018; Agrawal et al., 2016; Moradi et al., 2018; Nguyen et al., 2019).

---

[1]Our code is available here: https://github.com/DoubleBlindGithub/M3

Table 1: Overview of the M3 Benchmark. The M3 bencmark is comprised of six tasks — decompensation, length of stay, in-hospital (IH) mortality, long-term (LT) mortality, phenotyping, and readmission. For each task, we specify the number of available labeled training instances, the time(s) within the ICU stays at which predictions are made, the type of classification task, and the main metric used for evaluation. The M3 dataset also contains data from four modalities; we specify the dimensionality of each modality and the number of ICU stays containing data from it.

| Task | Training Instances | | | Prediction time(s) | Type | Main metric |
|------|---------|-------|--------|--------------------|------|-------------|
| | $|Train|$ | $|Val|$ | $|Test|$ | | | |
| Decomp. | 2495.8k | 360.3k | 523.1k | every hour | binary | aucpr |
| Length of Stay | 25.7k | 3673 | 5280 | 24 hour | multiclass | aucroc ovr |
| Mortality (IH) | 15.4k | 2209 | 3234 | 48 hour | binary | aucpr |
| Phenotyping | 30.7k | 4383 | 6278 | discharge time | multilabel | macro aucroc |
| Readmission | 27.4k | 3894 | 5659 | discharge time | multiclass | aucroc ovr |
| Mortality (LT) | 27.4k | 3894 | 5659 | discharge time | binary | aucpr |

| Modality | ICU stays | | | Dimensionality |
|----------|---------|-------|--------|----------------|
| | $|Train|$ | $|Val|$ | $|Test|$ | |
| time series | 30701 | 4383 | 6278 | 59 |
| clinical notes | 30507 | 4368 | 6247 | - |
| tabular inputs | 30701 | 4383 | 6278 | 6 |
| waveforms | 5567 | 686 | 1060 | 1 |

The clinical domain lends itself naturally to multi-modal learning, given the prevalence of data in different modalities, including physiological signals, clinical notes, medical images, tabular inputs and genome sequences. In this work, we propose the first multitask and multi-modal benchmark over this diverse clinical data.

## 3    THE MULTI-MODAL MULTI-TASK MIMIC BENCHMARK

Next, we introduce Multi-Modal Multi-Task MIMIC-III (M3), a dataset and benchmark for evaluating machine learning algorithms in healthcare. This dataset is derived from the MIMIC-III database and is comprised of data collected from over forty thousand patients. It contains data from four diverse modalities and defines six clinical tasks. It also proposes an evaluation framework to benchmark models and comes in an easy-to-use open source package.

Each patient in the dataset completes a number of stays in an ICU, with 13% of patients completing more than one stay. Over the course of each stay, the ICU collects measurements of vital signs and other clinical variables at irregular intervals, as well as clinical notes describing the state of the patient.

### 3.1    DESIGN PRINCIPLES

The goal of the M3 benchmark is to accelerate progress in applications of machine learning to healthcare. The design of this benchmark is guided by the principles of multi-modality — we collect data from diverse and unstructured modalities found in healthcare and that can be leveraged by modern deep learning algorithms — as well as task relevance, diversity, and evaluability — we choose diverse real-world clinical tasks that enhance model performance when solved jointly and that have well-defined success metrics. Finally, we are concerned with accessibility: in order to ensure that our benchmark can be widely adopted, we base it on datasets that can be obtained with minimum overhead and that satisfy privacy and legal requirements.

### 3.2    TASKS

The M3 benchmark includes 6 different clinical prediction tasks. Each task is performed at specific time points within the ICU stay of a patient.More details can be found in Table 1.

**In Hospital Mortality.**   We observe the first 48 hours of a patient's data, and then predict whether the patient will die by the end of their stay (Harutyunyan et al., 2019; Khadanga et al., 2020; Purushotham et al., 2017; Wang et al., 2020; Tang et al., 2020). Mortality is one of the major concerns for any ICU unit, with limiting mortality being an ultimate goal for most ICUs.

**Decompensation.**   Starting from the fifth hour of the stay, a prediction is made at every hour about whether the patient will die within the next 24 hours given all the data collected to that point. Unlike in the IHM task, predictions are made on an hourly basis rather than after a set amount of time and concern the next 24 hours rather than the entire stay. As such, this task may better reflect the changing landscape of available patient information.

**Length of Stay.**   We predict the total duration time from admission to discharge. This task could provide useful information for medical resources allocation and scheduling. We formulate this task as a multiclass classification problem with three classes/bins (0-3 days, 3-7 days, and longer than 7 days) using only data from the 24 hours of the stay.

**Phenotyping.**   A prediction of the patient's phenotype is made at discharge time. This is a multilabel classification task. The target label is derived from the billing code at a patients discharge, which we then convert to our 25 labels following the procedure from Harutyunyan et al. (2019).

**Readmission.**   We predict if another ICU stay will occur to the same patient after the discharge time of an ICU stay. Predicting readmission is useful to identify higher risk patients and minimize the waste of financial resources. We define this as a multiclass classification problem with 5 classes — readmission within 7, 7-30, 30-90, 90-365, and 365+ days or no readmission.

**Long-Term Mortality.**   For each ICU stay, we predict if the patient survives for more than 1 year after discharge. We frame this as a binary classification problem. Predicting long-term mortality is useful for assessing patients' well-being after discharge.

## 3.3   MODALITIES

Our dataset contains data from four diverse and unstructured modalities that are typically found in healthcare settings.

**Physiological Time Series.**   We select 59 temporal physiological variables from MIMIC-III, such as diastolic blood pressure, systolic blood pressure, oxygen saturation. The full list is in Table 8. These physiological variables are recorded irregularly, and they are important indicators of the patient's condition during the ICU stay.

**Clinical Notes.**   Clinical notes are written by clinicians and nurses during the ICU stay and usually summarize topics such as reasons for admission, details of treatment, nutrition, and the patients' respiratory conditions. These clinical notes are also temporal, and they are charted sporadically.

**Tabular Data.**   For every patient, we have access to their recorded sex, age, height, and weight upon entry to the ICU, the type of the ICU, and other tabular inputs. We consider only the initial values upon entry to the ICU. Some of these fields, such as weight, may fluctuate throughout the ICU stay, and are also part of the time Series data.

**Waveforms.**   Our dataset includes electrocardiogram (ECG) data from MIMIC Waveform (Moody, 2020). About 16% of patient stays have recorded waveform data.

## 3.4   DATASET AND BENCHMARK

We employed the preprocessing steps described in Harutyunyan et al. (2019), excluding ICU stays with missing events or missing length-of-stay and excluding patients younger than 18 years old because of the significant difference between adult and pediatric physiology.

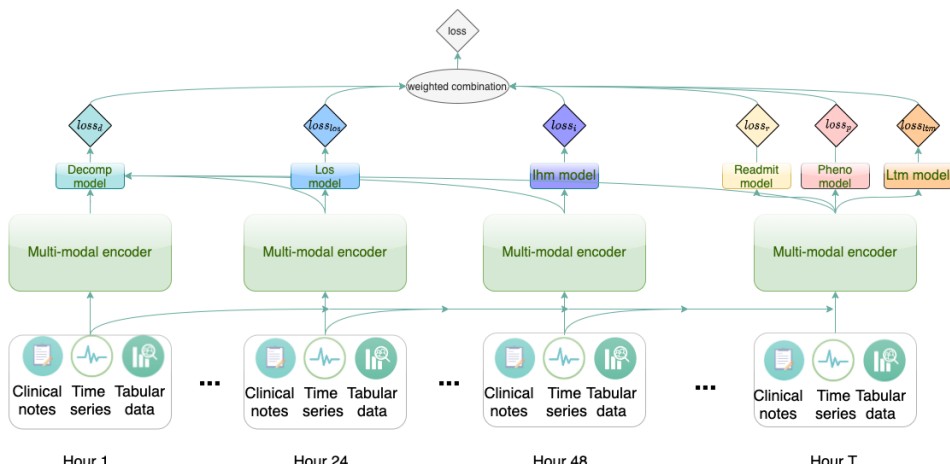

Figure 1: Multi-modal multi-task models for the M3 benchmark. We propose an architecture consisting of an encoder that outputs multi-modal embeddings based on text, time series, and tabular inputs as well as recurrent connections from earlier time steps (green horizontal lines). At each prediction time, these embeddings are used by task-specific components (e.g., "Decomp model" and others) to output predictions. Predictions are evaluated by task-specific losses and an overall multitask loss.

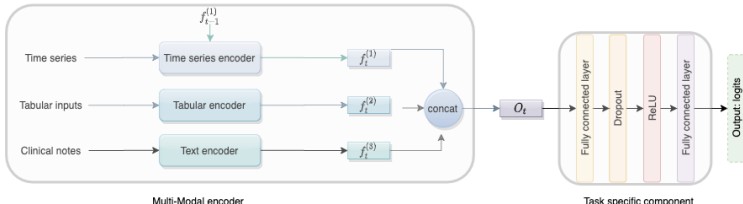

Figure 2: Multi-modal encoder and task-specific components. At each time step $t$, $f_t^{(1)}$ are the encoded time series features, $f_t^{(2)}$ are the encoded tabular inputs, and $f_t^{(3)}$ is the encoded clinical notes. The $O_t$ is a concatenated global embedding.

For binary classification problems, we define AUC-PR and AUC-ROC as the evaluation metrics. For multiclass classification problems, we define AUC-ROC-OVR (one versus rest), which compares the AUC of each class against the rest. We use macro AUC-ROC for the multilabel classification problem; it is the unweighted mean of AUC-ROC for each label.

# 4 MULTI-MODAL MULTI-TASK CLINICAL MODELS

As a demonstration of how our work can drive progress in clinical applications of machine learning, we propose a first set of multi-modal and multitask models for our new benchmark.

Our proposed architecture consists of an encoder that outputs multi-modal embeddings based on text, time series, and tabular inputs as well as recurrent connections from earlier time steps. The embeddings are fed to task-specific models that perform predictions.

## 4.1 ENCODERS

The multi-modal encoder is comprised of one child encoder per input modality (Figure 2). The multi-modal embedding is a concatenation of the outputs from each child encoder. We describe these below. Note that we do not use waveforms, as we found that we did not have enough inputs of this modality to improve model performance.

**Time Series.** Given a patient's ICU stay of length of T hours, we resample time series data with 1 hour interval and get $[\text{TS}_t]$ from $t = 1$ to $t = $ T. We impute missing values as described in modalities section. Our time series encoder is an LSTM (Hochreiter & Schmidhuber, 1997). The input $\text{TS}_t$ at time step $t$ is directly fed to the LSTM model along with the previous hidden state and cell state, and the next hidden state is our extracted feature, denoted by $f_t^1$. Formally, the feature extraction step is described as:

$$f_t^{(1)} = \text{LSTM}(\text{TS}_t, f_{t-1}^{(1)}) \tag{1}$$

For simplicity, cell states are omitted here.

**Clinical Notes** For each ICU stay, we have N clinical notes $[\text{Note}_{i=1}^{\text{N}}]$ charted sporadically, the charted time of these notes are $[\text{Time}(i)_{i=1}^{\text{N}}]$, where N is usually smaller than T. We use the text CNN proposed in Kim (2014) to extract text features. To create embeddings from $\text{N}_t$ notes collected at times $\text{Time}(i)$ preceding the current time step $t$ (i.e. $\text{Time}(i) \leq t$ for every $i \leq \text{N}_t$), we first extract features $[z_i]_{i=1}^{\text{N}_t}$ of each note by the text CNN separately, and then average the extracted features (Khadanga et al., 2020), assigning more weight to recent notes:

$$z_i = \text{TextCNN}(\text{Note}_i) \qquad \text{weight}(t,i) = \exp(-\lambda(t - \text{Time}(i))) \qquad f_t^{(2)} = \gamma \sum_{i=1}^{\text{N}_t} \text{weight}(t,i) \cdot z_i$$

Here, $\lambda$ is a scaling factor and $\gamma$ is a normalization term.

**Tabular Data** We have a total 6 tabular inputs that are non-temporal. To process the tabular inputs, we learn an embedding table for every categorical input dimension(de Brébisson et al., 2015). The embeddings for the tabular features are concatenated into one tabular embedding.

## 4.2 TASK-SPECIFIC COMPONENTS

Multi-modal embeddings are used as the input to task specific components, of which there is one per task. Each task specific component is composed of a fully connected layer with $h_t$ hidden units, a dropout layer, a ReLU activation, and an output layer matching the shape of that component's respective task. Each task specific component, regardless of architecture, has no explicit connection to any other task specific layer, but does share the same multi-modal embedding; thus the task specific components share the multi-modal encoder.

As shown in Figure 1, each task specific component has a task specific loss. To learn across all tasks simultaneously, we take the weighted sum of all the losses resulting to form the multi-task loss.

## 5 EXPERIMENTS

We now report performance on the M3 dataset. Our models achieve high performance levels and represent strong baselines for future work. In particular, we outperform previous state-of-the-art results that only rely on a subset of all tasks and modalities.

## 5.1 SETUP

Table 2: Implementation details for tabular encoder and task specific components.

(a) Tabular input features

| Tabular Feature | num values | Embed dim |
|:---:|:---:|:---:|
| ICU unit | 8 | 32 |
| Dbsource | 4 | 32 |
| sex | 4 | 32 |
| age | 100 | 32 |
| height | 100 | 32 |
| weight | 40 | 32 |

(b) Task specific components

| Task | hidden units($h_t$) | dropout prob($\alpha_t$) |
|:---:|:---:|:---:|
| Decomp. | 128 | 0.8 |
| Mortality (IH) | 108 | 0.85 |
| Pheno. | 512 | 0.1 |
| Len. of Stay | 32 | 0.8 |
| Mortality (LT) | 64 | 0.8 |
| Readm. | 128 | 0.1 |

**Encoders.** We used an 1-layer LSTM with 256 hidden units as the time series encoder. Clinical notes were mapped to Word2Vec embeddings (Mikolov et al., 2013). The Text CNN has three 1D kernels of size 2,3 and 4 with 128 filters each. We replaced all missing inputs with zeros.

**Task specific components.** We have 6 task specific components. Each task specific component has a linear layer with $h_t$ hidden units, followed by a dropout layer with dropout probability $\alpha_t$, a ReLU and an output layer. See Table 2b for details.

**Preprocessing.** The 17 clinical variables used by Harutyunyan et al. (2019) were normalized as in their paper. We imputed missing values during inference, taking the most recent value for the feature before the current time point. If there is no most recent value, then we impute based on predefined imputed values. For the additional time series features, we take the raw inputs and impute them in the same manner. Due to computational limits, we truncated each clinical note to at most 500 tokens and we limit the number of notes we can process per stay to 150. Inputs were zero-pad as necessary.

**Training** Every task used the cross entropy loss. To find multitask weights, we used uncertainty weighting (Cipolla et al., 2018), followed by manual search. We train with the Adam optimizer with a learning rate of 1e-4. and employ early stopping.

**Baselines** We compare to Harutyunyan et al. (2019), who report performance on physiological time series in a multi-task setting using LSTMs and channel-wise LSTMS. We also compared with Sheikhalishahi et al. (2020) who used BiLSTMs for time series data. For the text modality. We compared with Grnarova et al. (2016); Khadanga et al. (2020); Boag et al. (2018); Sheikhalishahi et al. (2020). Additionally, Khadanga et al. (2020) report a multi-modal result using time series and text. More details are in Table 4.

## 5.2 Main Results

Table 3 details the performance across every task for various input modalities and in both the single and multitask setting. Having additional modalities improves performance on every task except for length of stay, where it does not significantly change performance. Multitasking further increases performance on many tasks, such as in-hospital mortality, readmission, and long term mortality.

Table 4 shows the performance of models from different papers on the same tasks. Our models outperform both of our state-of-the-art baselines that only rely on a subset of all tasks and modalities.

Table 3: Results on all tasks by input modalities of time series (TS), clinical notes (Text) to tabular inputs (Tab). Tasks, in order, are Decompensation, In-Hospital Mortality, Phenotyping, Length of Stay, Long Term Mortality, and Readmission. Error bounds were computed by sampling with replacement 6000 samples 1000 times from the test set and computing metrics. Reported value is the median, bounds are based off of the 2.5 and 97.5 percentiles.

| Metrics | | Decomp. | Mortality (IH) | Pheno. | Len. of Stay | Mortality (LT) | Readm. |
|---|---|---|---|---|---|---|---|
| Modalities | Num Tasks | aucpr | aucpr | macro aucroc | aucroc ovr | aucpr | aucroc ovr |
| TS | ST | .299 ±.01 | .427 ±.053 | .788 ±.004 | .722 ±.012 | .291 ±.027 | .551 ±.02 |
| TS | MT | .280 ±.01 | .427 ±.053 | .77 ±.005 | .735 ±.012 | .366 ±.036 | .530 ±.02 |
| TS-Text | ST | .359 ±.01 | .580 ±.052 | .794 ±.005 | .751 ±.011 | .357 ±.030 | .700 ±.017 |
| TS-Text | MT | **.409** ±.01 | **.604** ±.05 | .773 ±.005 | **.754** ±.010 | .414 ±.032 | .696 ±.017 |
| TS-Text-Tab | ST | .404 ±.01 | .570 ±.051 | **.813** ±.004 | .737 ±.012 | .406 ±.031 | .695 ±.017 |
| TS-Text-Tab | MT | .408 ±.01 | .600 ±.05 | .812 ±.004 | .741 ±.011 | **.450**±.033 | **.708**±.016 |

## 6 Related Work

The paper by Harutyunyan et al. (2019) is the closest work to ours in the literature. They report the performance of multi-task models on a dataset derived from MIMIC-III that consists of 4 of our clinical prediction tasks and a subset of the physiological time series data that we use. Purushotham et al. (2017) introduced another benchmark covering tasks such as mortality prediction task, ICD code grouping task, and length-of-stay prediction with a different sets of time series features.

| Models | Decomp. | Mortality (IH) | Pheno. | Len. of Stay | Mortality (LT) | Readm. |
|---|---|---|---|---|---|---|
| | aucpr | aucpr | macro aucroc | aucroc ovr | aucpr | aucroc ovr |
| Single Modal Time Series | | | | | | |
| ts-LSTM | 0.299 | 0.427 | 0.788 | 0.722 | 0.291 | 0.551 |
| cw - LSTM | 0.326 | 0.486 | 0.788 | 0.774 | 0.341 | 0.594 |
| BiLSTM | 0.345 | 0.458 | 0.782 | 0.782 | 0.336 | 0.656 |
| Single Modal - Text | | | | | | |
| LSTM | NA | NA | 0.546 | NA | NA | NA |
| CNN | 0.312 | 0.517 | 0.541 | 0.711 | 0.269 | 0.563 |
| Multi Modal - TS + Text | | | | | | |
| ts-LSTM + CNN | 0.359 | 0.580 | 0.794 | 0.751 | 0.357 | 0.700 |
| cw-LSTM + CNN | 0.387 | 0.600 | 0.790 | 0.760 | 0.400 | 0.692 |
| Multi Modal - TS + Text + Tab | | | | | | |
| ts-LSTM + CNN + Tab | 0.404 | 0.570 | 0.813 | 0.737 | 0.406 | 0.695 |

Table 4: Comparing Single modal baselines and increasing modalities. Harutyunyan et al. (2019) use time series LSTMs(ts-LSTM) and channelwise LSTMs(cw-LSTMs) in their work. Sheikhalishahi et al. (2020) achieve their best performance with BiLSTMs. For text modality. LSTM text classifier are reported to achieve state-of-the-art performance on phenotyping task (Grnarova et al., 2016). For all other tasks, the best model we found in our literature search is Text CNN (Khadanga et al., 2020; Boag et al., 2018; Sheikhalishahi et al., 2020). Khadanga et al. (2020) reported a TS+Text Multi Modal model ts-LSTM+CNN.

Khadanga et al. (2020) are the first to consider the multi-modal setting, specifically clinical notes and time series data. They focus on three of our clinical prediction tasks (decompensation, length of stay, and in-hospital mortality), but do not perform multi-task learning. Our paper improves over both Harutyunyan et al. (2019) and Khadanga et al. (2020). We create a benchmark that combines both multiple modalities (including two modalities not found in Khadanga et al. (2020)) and multiple tasks (including two new tasks not present in earlier works).

## 7 DISCUSSION

We release our benchmark in an easy-to-use open source package. This resource may be used to improve architectures for biomedical multi-modal and multi-task learning, as well as to explore research directions in causality, generative models, fairness, and other areas.

**Future Research Directions.** Within the field of causal inference, our dataset can be used to study ways of inferring latent causal factors from unstructured modalities such as images. It may also drive experiments in new generative model architectures and ways in which generative models can address practical problems such as missing data inputs or the estimation of uncertainties in a multi-task regime. Our diverse dataset may offer insight into questions of fairness and privacy, such as on the extent to which medical images and clinical notes reveal sensitive patient information such (e.g., race or gender), and on how to address this leakage. We hope that our benchmark will serve as a resource to a broad subset of the machine learning community.

## 8 CONCLUSION

In this paper, we argued that developing new machine learning models in healthcare that effectively leverage diverse modalities — such radiology images, clinical notes, and sensor data — and that can simultaneously solve multiple tasks — e.g., estimating disease risk, predicting readmission — has the potential to significantly impact applications in the clinical domain. We proposed the first benchmark for the healthcare domain that focuses on multi-modal and multi-task learning and new models that outperform previous state-of-the-art methods that only rely on a subset of tasks or modalities.

Our results highlight the importance of multi-modal and multitask learning in clinical settings. More generally, we envision our work being a general resource that will help accelerate research in applying machine learning to healthcare.

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

# A APPENDIX

## A.1 FULL PERFORMANCE PER TASK

Table 5: Length of stay

|  | AUCROC ovr | AUCROC ovo |
| --- | --- | --- |
| TS-ST (ours) | 0.722 (0.711, 0.734) | 0.692 (0.682, 0.703) |
| TS-MT (ours) | 0.735 (0.723, 0.746) | 0.700 (0.690, 0.710) |
| Ts-Text-ST (ours) | 0.751 (0.740, 0.762) | 0.730 (0.719, 0.741) |
| TS-Text-MT (ours) | **0.754 (0.743, 0.765)** | **0.734 (0.723, 0.745)** |
| TS-Text-Tab-ST (ours) | 0.737 (0.725, 0.748) | 0.713 (0.702, 0.724) |
| TS-Text-Tab-MT (ours) | 0.741 (0.730, 0.752) | 0.719 (0.709, 0.730) |

Table 6: Long term mortality

|  | AUCROC | AUCPR |
| --- | --- | --- |
| TS-ST (ours) | 0.659 (0.640, 0.677) | 0.291 (0.266, 0.318) |
| TS-MT (ours) | 0.736 (0.720, 0.753) | 0.366 (0.337, 0.396) |
| TS-Text-ST (ours) | 0.701 (0.683, 0.718) | 0.357 (0.328, 0.388) |
| TS-Text-MT (ours) | **0.760 (0.744, 0.775)** | **0.414 (0.382, 0.445)** |
| TS-Text-Tab-ST (ours) | 0.744 (0.728, 0.760) | 0.406 (0.375, 0.438) |
| TS-Text-Tab-MT (ours) | 0.784 (0.768, 0.798) | 0.450 (0.417, 0.482) |

Table 7: Readmission

|  | AUCROC ovr | AUCROC ovo |
| --- | --- | --- |
| TS-ST (ours) | 0.551 (0.532, 0.571) | 0.532 (0.517, 0.546) |
| TS-MT (ours) | 0.530 (0.510, 0.550) | 0.524 (0.508, 0.540) |
| TS-Text-ST (ours) | 0.700 (0.683, 0.717) | 0.600 (0.588, 0.612) |
| TS-Text-MT (ours) | 0.696 (0.679, 0.713) | 0.603 (0.590, 0.616) |
| TS-Text-Tab-ST (ours) | 0.695 (0.678, 0.712) | 0.603 (0.589, 0.617) |
| TS-Text-Tab-MT (ours) | **0.708 (0.692, 0.725)** | **0.605 (0.594, 0.617)** |

## A.2 COMPLETE PHYSIOLOGICAL TIME SERIES VARIABLES

| Time series var |
| --- |
| Alanine aminotransferase |
| Albumin |
| Alkaline phosphate |
| Anion gap |
| Asparate aminotransferase |
| Basophils |
| Bicarbonate |
| Bilirubin |
| ,Blood urea nitrogen(BUN) |
| CO2 (ETCO2; PCO2; etc.) |
| Calcium |
| Calcium |
| Ionized |
| Calcium ionized |
| Capillary refill rate |
| Chloride,Cholesterol |
| Creatinine |
| Diastolic blood pressure |
| Eosinophils |
| Fraction inspired oxygen |
| Glascow coma scale eye opening |
| Glascow coma scale motor response |
| Glascow coma scale total |
| Glascow coma scale verbal response |
| Glucose |
| Heart Rate |
| Hematocrit |
| Hemoglobin |
| Lactate |
| Lactate dehydrogenase |
| Lactic acid,Lymphocytes |
| Magnesium |
| Mean blood pressure |
| Mean corpuscular hemoglobin |
| Mean corpuscular hemoglobin concentration |
| Mean corpuscular volume |
| Monocytes |
| Neutrophils |
| Oxygen saturation |
| Partial pressure of carbon dioxide |
| Partial pressure of oxygen |
| Partial thromboplastin time |
| Peak inspiratory pressure |
| Phosphate |
| Platelets |
| Positive end-expiratory pressure |
| Potassium |
| Prothrombin time |
| Red blood cell count |
| Respiratory rate |
| Sodium |
| Systolic blood pressure |
| Temperature |
| Urine output |
| Weight |
| White blood cell count |
| pH |

Table 8: List of all used Time series vars

Table 9: Full results with all metrics

| Modalities | Num Tasks | Decomp. aucroc/ aucpr | Mortality (IH) aucroc/aucpr | Pheno. aucroc macro/micro | Len. of Stay aucroc ovr/ovo | Mortality (LT) aucroc/aucpr | Readm. aucroc ovr/ovo |
|---|---|---|---|---|---|---|---|
| Time Series | ST | .896 ±.003/.299 ±.01 | .850 ±.023/.427 ±.053 | .788 ±.004/.837 ±.004 | .722 ±.012/.692 ±.011 | .659 ±.019/.291 ±.027 | .551 ±.02/.532 ±.015 |
| Time Series | MT | .892 ±.003/ .280 ±.01 | .858 ±.020/.427 ±.053 | .77 ±.005/.82 ±.005 | .735 ±.012/.700 ±.010 | .736 ±.016/.366 ±.036 | .530 ±.02/.524 ±.016 |
| Time Series-Text | ST | .925 ±.002/.359 ±.01 | .880 ±.018/.580 ±.052 | .794 ±.005/.840 ±.004 | .751 ±.011/.730 ±.011 | .701 ±.018/.357 ±.030 | .700 ±.017/.600 ±.012 |
| Time Series-Text | MT | .926 ±.003/.409 ±.01 | **.890** ±.018/**.604** ±.05 | .773 ±.005/.825 ±.006 | **.754** ±.010/**.734** ±.011 | .760 ±.016/.414 ±.032 | .696 ±.017/.603 ±.013 |
| Time Series-Text-Tabular | ST | .927 ±.003/.404 ±.01 | .877 ±.017/.570 ±.051 | **.813** ±.004/**.855** ±.004 | .737 ±.012/.713 ±.011 | .744 ±.016/.406 ±.031 | .695 ±.017/.603 ±.014 |
| Time Series-Text-Tabular | MT | **.934** ±.003 / **.408** ±.01 | .887 ±.016/.600 ±.05 | .812 ±.004/.854 ±.003 | .741 ±.011/.719 ±.011 | **.784** ±.016/**.450** ±0.33 | **.708** ±.016/**.605** ±0.11 |