# OpenReview forum: "A Multi-Modal and Multitask Benchmark in the Clinical Domain"
_ICLR.cc/2021/Conference — Reject_

### Official Review · AnonReviewer4 · 2020-10-27
**A Work on Clinical Prediction with A New Setting**

**Rating:** 5
**Confidence:** 5

**Review:**

This paper defines a new task for clinical data by combing multi-modal and multi-task settings into one task. It collects a dataset called M3 as the benchmark for the multi-modal and multi-task benchmark in the clinical domain. The dataset has 6 prediction tasks, i.e., in-hospital mortality, decompensation, length of stay, phenotyping, readmission, and long-term mortality, and it has 4 modalities, i.e., physiological time series, clinical notes, tabular data, and waveforms. Specifically, this paper also provides a multi-modal multi-task model where the time series data are encoded by LSTM, clinical notes are encoded by text CNN and tabular data are also encoded by existing methods.  In experiments, the authors conduct an ablation study and compare the proposed method with the method of Harutyunyan et al. and Khadanga et al.

Quality: The overall quality of the paper is marginally below the acceptance threshold. The problem definition, data collection, and model design are reasonable. However, I am concerned about the paper presentation and the experiment design. The authors need to compare the proposed multi-modal multi-task model with state-of-the-art single-modality models.

Clarity: The presentation of the paper is easy to follow, but the structure of the presentation may need to be improved.

Originality: The collected dataset is also new because of the new setting. But in terms of model design, the feature embeddings are learned by existing methods.

Significance of This Work: The direction of this work is significant and worth being paid attention to. Considering multi-modal multi-task settings in the clinical domain is useful for the development in this area.

Pros:
1. The multi-modal multi-task setting is interesting and important for future related research. This paper provides a new direction for moving machine learning forward in the clinical domain.
2. Collecting data from MIMIC-III is a reasonable choice for creating a multi-modal multi-task dataset. It provides a solution on how to create a new dataset for clinical prediction task with new settings.
3. The model design is reasonable. I think the authors choose the right frameworks for dealing with different modalities.

Despite this, I am concerned about the presentation and experimental design of the paper, which are summarized as the cons as follows.

Cons:
1. The paper structure could be improved by incorporating Section 2 “background” and Section 6 “related work” together. The mentioned work in Section 6 is related to “machine learning in the clinical domain” in Section 2.
2. There are other forms of multi-modal learning in the clinical domain, and the authors should take them into consideration and discuss them in the background. For example, Moradi et al. and Nguyen et al. have some work about the text and image multi-modal learning in the clinical domain.
3. As for the experiment, Table 3 is useful in showing the model design, but I think the result presentation and experiment design in Table 4 can be improved. The authors could provide how the choices of encoders can influence task performance if there are only two baselines. This can help the readers see what’s the potential of doing research on this dataset and directions.  For example, the method of Harutyunyan et al. only uses the time series modality in Table 4, but it can be used as the encoder to process time series in the proposed model design, and maybe the authors should consider this setting as a baseline. To sum up, I think the comparison experiment in Table 4 is not compelling enough to illustrate the model design.
4. There are many existing works using single modality data towards the six tasks. The authors should compare those models on a single modality to demonstrate improvement when incorporating a multi-modal multi-task dataset.

Some typos: In the conclusion section, “he first benchmark” should be “the first benchmark”

References
Moradi, M., Madani, A., Gur, Y., Guo, Y., & Syeda-Mahmood, T. (2018, September). Bimodal network architectures for automatic generation of image annotation from text. In International Conference on Medical Image Computing and Computer-Assisted Intervention (pp. 449-456). Springer, Cham.
Nguyen, B. D., Do, T. T., Nguyen, B. X., Do, T., Tjiputra, E., & Tran, Q. D. (2019, October). Overcoming data limitation in medical visual question answering. In International Conference on Medical Image Computing and Computer-Assisted Intervention (pp. 522-530). Springer, Cham.

---

> ### Author Response · Authors · 2020-11-24
> **Clarifications and updates to table 4**
>
> We appreciate the feedback on the presentation of table 4. We apologize for the confusion and reorganized our results in comparison to baselines. As mentioned in the review, there are several works that utilized a single modality for some subset of the six tasks. For example, Sheikhalishahi et al. used Bi-LSTM to achieve the state-of-the-art performance on the in-hospital mortality task using only time-series modality. However, due to reasons such as different datasets, different cohort selection, and different preprocessing steps, we cannot directly compare to the numbers they report.  Most models found were not directly comparable, so, we re-implemented the models used in these previous works and ran experiments with standardized preprocessing for a fair comparison. We listed results from LSTM, channel-wise LSTM, Bi-LSTM that we found to be either state-of-the-art models or common baselines.   Table 1 in this response shows performance metrics per task with each of these models as the Time Series encoder. All the following tables can be viewed together under table 4 in the paper.
>
>
> | Model | Decomp | Mortality(IH) | Pheno. | Len. of Stay | Mortality(LT) | Readm |
> |--|--|--|--|--|--|--|
> |Metric|AUCPR|AUCPR| macro AUCROC| AUCROC ovr| AUCPR | AUCROC ovr|
> |LSTM | 0.299 | 0.427 | 0.788 | 0.722 | 0.291 | 0.551|
> |cw-LSTM |0.3262 | 0.4859 | 0.788 | 0.774 | 0.341 | 0.594|
> |bi-LSTM |0.345 | 0.458 | 0.782 | 0.782 | 0.336 | 0.656|
>
>
>
> Similarly, we found that Text CNN and LSTM are state-of-the-art models using only the text modality.  Due to the high computational costs of text LSTM, we only ran experiments on phenotyping task using LSTM, and to our knowledge, phenotyping is the only task that the  SOTA performance is achieved with LSTM.  For all other tasks, the Text CNN outperforms the LSTM. We list the performance of using these models with only text in table 2 in this response.
>
>
>
>
> | Model | Decomp | Mortality(IH) | Pheno. | Len. of Stay | Mortality(LT) | Readm |
> |--|--|--|--|--|--|--|
> |Metric|AUCPR|AUCPR| macro AUCROC| AUCROC ovr| AUCPR | AUCROC ovr|
> |LSTM | NA | NA | 0.546 | NA | NA | NA|
> |Text CNN |0.312 | 0.517 |0.541 | 0.711| 0.269 | 0.563|
>
>
>
> Khadanga et al. also attempted to combine time-series data and text data, but they only tried on 3 tasks out of 6 and tested their performance on a smaller dataset, and their implementation leaked some future notes. We reimplemented their model, which uses a standard LSTM for time series data and a text CNN for text data while also making sure not to leak any future notes. In addition, we also implemented a channel-wise LSTM for time series data and a text CNN for text data. The results of our implementation on 6 tasks  are in the following table.
>
>
> | Model | Decomp | Mortality(IH) | Pheno. | Len. of Stay | Mortality(LT) | Readm |
> |--|--|--|--|--|--|--|
> |Metric|AUCPR|AUCPR| macro AUCROC| AUCROC ovr| AUCPR | AUCROC ovr|
> |LSTM + CNN |0.359 | 0.58 | 0.794 | 0.751 | 0.357 | 0.700 |
> |cw-LSTM + CNN |0.387 | 0.6 | 0.79 | 0.76 | 0.40 | 0.692
>
>
> Finally, we present our performance with all modalities to demonstrate that multimodal learning may lead to performance gains in some tasks.
>
>
> | Model | Decomp | Mortality(IH) | Pheno. | Len. of Stay | Mortality(LT) | Readm |
> |--|--|--|--|--|--|--|
> |Metric|AUCPR|AUCPR| macro AUCROC| AUCROC ovr| AUCPR | AUCROC ovr|
> |Text + Time-series +Tab(ours) |0.404 | 0.570 |0.813 | 0.737| 0.406 | 0.695|
>
> Thank you for all your feedback!
>
> Reference:
>
> [1] [**Benchmarking machine learning models on multi-center eICU critical care dataset**](https://journals.plos.org/plosone/article?id=10.1371/journal.pone.0235424 "Back to original article")
>
> Sheikhalishahi S, Balaraman V, Osmani V (2020) Benchmarking machine learning models on multi-center eICU critical care dataset. PLOS ONE 15(7): e0235424. [https://doi.org/10.1371/journal.pone.0235424](https://doi.org/10.1371/journal.pone.0235424)
>
> [2] Grnarova, Paulina, et al. "Neural document embeddings for intensive care patient mortality prediction." _arXiv preprint arXiv:1612.00467_ (2016).
>
> [3] Khadanga, Swaraj, et al. "Using Clinical Notes with Time Series Data for ICU Management." _arXiv preprint arXiv:1909.09702_ (2019).
>
> [4] Boag, Willie, et al. "What’s in a note? unpacking predictive value in clinical note representations." _AMIA Summits on Translational Science Proceedings_ 2018 (2018): 26.

---

### Official Review · AnonReviewer2 · 2020-10-28
**bencahmark MIMIC-III data but not robust evaluation for the proposed model**

**Rating:** 5
**Confidence:** 4

**Review:**

There are two objectives for this work:
1. A new MIMICIII benchmark data that supports multi tasks and multi modalities
2. A proposed multi-modal multi-task AI model

For the first objective, the data supports 6 clinical classification tasks (which were proposed before in a earlier MIMICIII benchmark) but also supports 4 different modalities: physiological time series (the standard modality that used in the prior benchmark), clinical notes, baseline data, and waveform (although not being used in the method/experiments). For the second objective, a modality-specific embedding model was applied to each modality independently and then an aggregation of all embeddings was used as an input to a task-specific predictive model. The weighted linear combination of losses over all tasks was used for the multi-task settings.



I have the following concerns/comments:
- MIMIC-IV was released long time back and MIMIC-V was just released a few months back. Why the authors benchmarking MIMIC-III and not the recent versions?
- The authors have mentioned "We also propose an evaluation framework to benchmark models on this dataset.". I could not see that in the paper
- "We release M3 and our models as an easy-to-use open-source package for the research community". Where is the link? I can not evaluate "easy-to-use" open source package.
- The number of stays in Table 1, is that when all modalities are available or the number when any modality is available?
- I hope in the open-source package (which is not provided) if the clinical notes are processed so that it can be used by other researchers
- There is no clear description about how waveforms are processed.
- "we resample time series data with 1 hour", what would be the case if multiple data points are available within 1 hour? Do the authors average them or take the recent value?
- For tabular data "To process the tabular inputs, we learn an embedding table for every categorical input dimension". why do not learn embedding use all of them instead of learning embedding for each feature independently. The correlation among these features might not be captured.
- I give the authors a credit of using weighted combination of losses using the uncertainty of each task
- In the baselines section, the authors mentioned " We replicate their test set to compare to their results", what does it mean? What would be the case if the part of the test data in the "Khadanga et al. (2020)" is part of the training in this work
- "Having additional modalities improves performance on every task except for length of stay," The same also applies for in-hospital mortality
- The presentation of Table 3 could be significantly improve if you start with the entire model (all modalities) and compare it to counter-part single task. Then in a separate table, you perform ablation analysis to see which modality is important.
- Table 4 is confusing. Maybe another representation could be a figure where in x-axis the metric for the baseline and y-axis is the metric for the proposed model and each do represents one comparison. For example, the first part of the Table could be a dot (model) where y-axis is 0.408 and x-axis is 0.344, which compares the proposed model versus the baseline model. The legend of the figure spells out the task name (and any other info to be added).
- The numbers in Table 4 do not match, for example the second part of the table shows that the proposed models has 0.417 AUCPR, where does that number come from? It is not in Table 3.
- The discussion and future work is not quite related to the paper, for example, it was mentioned about isng images, but images is not one of the modalities in the proposed benchmark. Do you plan to add images to he benchmark? Does MIMIC has images?
- There is no comparison versus other multi-modal multi-task baselines
- Change "We proposed he first benchmark" to "We proposed the first benchmark"
- Change "Given a patient’s ICU stay of length of stay T hours" to "Given a patient’s ICU stay of length T hours"
- There is no section about how the hyperparameters were optimized

---

> ### Author Response · Authors · 2020-11-24
> **Response to reviewer 2**
>
> Dear Reviewer, Thank you for your feedback. We appreciate your feedback on structure and changed the presentation of table 4 to aid in clarity. There are many questions/comments listed and we will answer them point by point, and we hope this will address your questions and concerns.
>
> -  Q:  MIMIC-IV was released long time back...?
> - R: To our knowledge, there is no MIMIC-V at this time. MIMIC-IV was made publicly available on August 13th of this year. MIMIC-IV is currently on version 0.4, so there may be some schema changes in the future, and more data from MIMIC-IV is coming in future, but MIMIC-III should not receive any major changes of this sort and so was a stable and still current at time of development dataset.
>
> -  Q: The authors have mentioned "We also propose an evaluation framework...".
> - R: The evaluation framework we describe is the way to construct a model, the tasks to train, and some initial baselines for those tasks. We also provide a software platform to allow for researchers to allow researchers to expand upon the work.
>
> -  Q: "We release M3 and our models as an easy-to-use open-source package...
> - R: The link to our implementation is provided in Footnote at bottom of page 2, link provided again for convenience: https://github.com/DoubleBlindGithub/M3
>
> -  Q: The number of stays in Table 1, is that...?
> - R: The stays column is the number of stays that contain that modality.
>
> -  Q: I hope in the open-source package …
> - R:  Text processing scripts in directory src/scripts in the repository
>
> -   Q: There is no clear description about how waveforms are processed.
> - R: From our experiment, though we found that waveform data did not improve the performance in a significant way, we think it could still be useful for future extension. Therefore, we focus more on other modalities which are more important in this paper and leave the process of waveform data in our codebase for reference. Most of the data processing was done as  described in MIMIC-waveform(https://physionet.org/content/mimic3wdb/1.0/)[fix citation]. We then selected for various leads(I,II, V, etc.) of 12 lead ECGs when they were available. There is more waveform data present, but due to size and computational constraints we did not include it in further analysis.
>
> -  Q:  "we resample time series data with 1 hour", what would be the case ...?
> - R: In the case if multiple data points are available within 1 hour, we take the most recent value
>
> -   Q: For tabular data "To process the tabular inputs, we learn an embedding table for every categorical input dimension"...
> - R: The correlation among these features might not be captured in the tabular embedding. However, we choose to have seperate embeddings per category because otherwise the size of the table would lead to many unique embeddings, with a large percentage of joint embeddings having few samples. We could try to mitigate this by pairing the categorical variables instead of having an embedding per each, but that is outside the current scope of the paper. Relationships between different features may theoretically still be captured down stream, at a task specific component for example. How to best process these sorts of tabular embeddings is still an open area of research though.
>
>
>
> -  Q: In the baselines section, the authors mentioned ...
> - R: Khadanga et al ended up doing some additional filtering to their train and test sets, so their training and test sets are strictly subsets of the training and test splits we use. This led to us replicating their test set in order to compare to them. However, we have reworked table 4 so that all models are using the same splits to avoid any confusion.
>
> -  Q:  Having additional modalities improves performance ...
> - R: The same also applies for in-hospital mortality For In hospital mortality, we observe a aucpr of .427 using only the time series modality. This improves to .604 when incorporating the text modality. Then using time series, text, and tabular data, we observe an aucpr of .600. When going from time series to ts+text, we do see an increase in performance, but when adding tabular data in we don’t see much of a change.
>
> -  Q: The numbers in Table 4 do not match…
> - R: Apologies for the confusion, the second and fourth tables use the Khadanga et al test set, and so they are different from the test set used in table 3.  We have reworked how table 4 is structured so that they are all on the same test set to avoid this confusion.
>
> -  Q: The discussion and future work is not quite related to the paper...?
> - R: We would like to extend this work to MIMIC-IV in the future, and MIMIC-IV contains a mapping to MIMIC-CXR, a database of radiography data(containing images and notes about those images). This would allow us to incorporate an image modality.
>
> -   Q: There is no comparison versus ..
> - R:  To our knowledge, we are the first multi-modal and multi task work in this field.

---

### Official Review · AnonReviewer3 · 2020-10-29
**Expands Prior Work**

**Rating:** 5
**Confidence:** 4

**Review:**

################################

Summary:

This paper discusses the Multi-Modal Multi-Task MIMIC-III (M3) dataset and benchmark, which extends previous efforts in this space to provide a benchmark on the MIMIC-III dataset. In particular, this work considers the inclusion of multiple modalities, including time series, clinical notes, ECG waveforms, and tabular input. It also defines six clinical tasks, some of which overlap with existing efforts and others which appear to be new.

################################

Reasons for score:

Overall, I lean toward reject. This appears to be a simple extension of existing work in the area. While in many ways it's a meaningful contribution to this area it also rests at the boundary of the scope for the CFP. Either alone might not be grounds for rejection, but together they likely put this work below the acceptance threshold.

################################

Strengths:

- Unifying existing work. In looking to the repository (https://github.com/DoubleBlindGithub/M3), it appears that the incorporates existing benchmarks by Harutyutyan et al. This approach of building on existing benchmarks rather than reinventing the wheel seems like a great start toward unifying work in this space.

- Multi-Modal. While prior works note extensibility to include additional modalities, they tend to focus on a single modality and merging multiple modalities in a composable manner may require some additional work. This paves the way to provide additional multi-modal work in this area.

- Timely domain. There has been increasing interest in healthcare due to the current COVID-19 pandemic. The availability of good benchmarks stands to channel that interest and energy into improvements that matter.

################################

Weaknesses:

- Clinical actionability / best practices. In cited works, particularly Wang et al., there is a presence of a "gap" introduced in the tasks to discourage overly easy predictions from the data. This work appears to remove those gaps and thus benchmark results may be unusually high and omit what appears to be a best practice for work in this space.

- Unclear takeaway. The models that are reported support a claim that additional information (in the form of additional modalities) are able to improve performance in general, but do not clarify the importance of multi-task in this setting. Expanding the discussion may help elucidate the insights that should be taken away from this work.

- Missing reference. Tang et al.'s "Democratizing EHR analyses with FIDDLE: a flexible data-driven preprocessing pipeline for structured clinical data" (https://academic.oup.com/jamia/advance-article/doi/10.1093/jamia/ocaa139/5920826) should be added as related work.

################################

Questions:

- Can additional context be provided for this work as it relates to the scope of ICLR?

- Can you clarify the takeaways from the results of this work, or clarify if the paper serves primarily to introduce the benchmark?

---

> ### Author Response · Authors · 2020-11-24
> **Temporal Gap and other clarifications**
>
> Dear Reviewer, Thank you for your comments and feedback. And here is our responses to the questions and concerns.
>
>
>
> 1. Temporal Gap - To our knowledge, there is no widely accepted way to determine what temporal gap/lead time should be used(these terms are used interchangeably in the literature). Papers like the one the reviewer pointed out use a temporal gap of zero(Tang et al., noted in the supplementary section page 13), while follow up papers by the Mimic-extract authors use temporal gaps that vary between 2-12 hours(McDermott et al). We did experiment with different temporal gaps and saw changes in performance in the in-hospital mortality task. For example, a temporal gap of 6 hours would decrease the aucpr of our model from 0.6 -> 0.576. It is unclear how much of this change should be attributed to the decrease in data(~14% decrease) as opposed to stopping any leakage. For context, 139 (0.9%) of stays that expire are within a 6 hour gap.  It is difficult to determine when exactly data is being leaked, so it is difficult to evaluate different temporal gap/lead time lengths. To summarize, we believe more research is required to determine an adequate gap, and we have seen work both include and not include these temporal gaps. We choose to work with the most data possible  since we wish to evaluate these models under ideal circumstances. Moving this into a clinical setting would likely require some gap time, although that time may depend on the specific setting.  We do include a setting to allow users to specify a lead time for the tasks that would require them(the In hospital mortality task and the Length of stay task).
>
> 2. Multitasking - We observed that in some tasks, training using multitasking resulted in a performance increase. We believe multitasking to be a useful consideration when training. In addition, some recent work by McDermott et al. shows that multitasking may help with smaller datasets. As such, we think multitasking holds promise in this setting.
>
> 3. Relevancy - In the non-exhaustive list of topics, the CPF calls for “applications in …, computational biology, or any other field” and “implementation issues, parallelization, software platforms, …”. This work tackles both an application aspect(how would a multimodal and multi task model in this setting look), and provides a software platform to allow researchers to build off of. As such, we believe this work fits into the scope of ICLR.
>
> 4. One key takeaway from this is that we observed multimodal learning and multitask learning is a very promising path to build practically useful predictive models in the healthcare domain. This is a benchmark paper, and another contribution of this work besides presenting a benchmark is that we provided an easy-to-use open-source package of preprocessing complex multi-modal clinical data for the community.
>
> ### Supplemental
>
> 1. [Some data for reference] 14% of all our In hospital Mortality data within a temporal gap of 6 hours. Within a 6 hour gap, only 139 patients expire out of 2064(6.7%) total patients that expire. Within a 12 hour window, 242 patients expire out of total expirations(11.7%). The stays that expire within a 6 hour gap(139) are only ~0.9% of all stays in our training set. Incorporating a 6 hour gap decreases performance a bit(as would be expected, unsure if this is because less data or because of actual leakage)(aucpr 0.600 -> 0.576). Papers like Ren et al. explore this a bit.
>
> 2. Ren et al. conduct some analysis on predicting respiratory decompensation. This is not a task we look at, but their analysis takes into account both the size of the window for prediction and the lead time(gap between observation and prediction).  For all models, larger lead time -> worse performance. The delta between performance at low lead times and high lead times is different depending on models however. See figure 4a in Ren et al. W is the window time(how much data could they see at prediction time), L is the lead time(temporal gap). Different models have different performance changes wrt lead time. The interaction of window size, lead times, and model choice are still unclear.
>
> ### Reference
> 1. Shengpu Tang, Parmida Davarmanesh, Yanmeng Song, Danai Koutra, Michael W Sjoding, Jenna Wiens, Democratizing EHR analyses with FIDDLE: a flexible data-driven preprocessing pipeline for structured clinical data, _Journal of the American Medical Informatics Association_, , ocaa139, [https://doi.org/10.1093/jamia/ocaa139](https://doi.org/10.1093/jamia/ocaa139)
> 2. McDermott, Matthew, et al. "A Comprehensive Evaluation of Multi-task Learning and Multi-task Pre-training on EHR Time-series Data." _arXiv preprint arXiv:2007.10185_ (2020).
> 3. O. Ren _et al_., "Predicting and Understanding Unexpected Respiratory Decompensation in Critical Care Using Sparse and Heterogeneous Clinical Data," _2018 IEEE International Conference on Healthcare Informatics (ICHI)_, New York, NY, 2018, pp. 144-151, doi: 10.1109/ICHI.2018.00024.

---

### Decision · Program_Chairs · 2021-01-07
**Final Decision**

**Decision:**

Reject

**Comment:**

The paper has two contributions. A novel benchmark for clinical multi-modal multi-task learning based on the already released MIMIC III and a multi-modal multi-task machine learning model. While the paper does show value in providing a curated benchmark and combining/unifying existing approaches to a timely problem, the reviewers agree that the paper provides insufficient novelty to warrant publication.